# Digital physiotherapy assessment vs conventional face-to-face physiotherapy assessment of patients with musculoskeletal disorders: A systematic review

Susanne Bernhardsson[1,2]*, Anette Larsson[1,3,4], Anna Bergenheim[1,2], Chan-Mei Ho-Henriksson[1,2,5], Annika Ekhammar[1,2,6], Elvira Lange[1,2,3], Maria E. H. Larsson[1,2,7], Lena Nordeman[1,2], Karin S. Samsson[1,2,8], Lena Bornhöft[1,2,9]

1 Research, Education, Development and Innovation, Primary Health Care, Region Västra Götaland, Vänersborg, Sweden, 2 Unit of Physiotherapy, Department of Health and Rehabilitation, Institute of Neuroscience and Physiology, Sahlgrenska Academy, University of Gothenburg, Gothenburg, Sweden, 3 Department of General Practice / Family Medicine, School of Public Health and Community Medicine, Institute of Medicine, Sahlgrenska Academy, University of Gothenburg, Gothenburg, Sweden, 4 Närhälsan Herrljunga Rehabilitation Clinic, Primary Care Rehabilitation, Region Västra Götaland, Herrljunga, Sweden, 5 Närhälsan Lidköping Rehabilitation Clinic, Primary Care Rehabilitation, Region Västra Götaland, Lidköping, Sweden, 6 Närhälsan Eriksberg Rehabilitation Clinic, Primary Care Rehabilitation, Region Västra Götaland, Gothenburg, Sweden, 7 Centre for Clinical Research and Education, Region Värmland, Karlstad, Sweden, 8 Capio Ortho Center Gothenburg, Gothenburg, Sweden, 9 Närhälsan Torslanda Rehabilitation Clinic, Primary Care Rehabilitation, Region Västra Götaland, Gothenburg, Sweden

* susanne.bernhardsson@gu.se

## Abstract

### Background

This systematic review aimed to assess the certainty of evidence for digital versus conventional, face-to-face physiotherapy assessment of musculoskeletal disorders, concerning validity, reliability, feasibility, patient satisfaction, physiotherapist satisfaction, adverse events, clinical management, and cost-effectiveness.

### Methods

Eligibility criteria: Original studies comparing digital physiotherapy assessment with face-to-face physiotherapy assessment of musculoskeletal disorders. Systematic database searches were performed in May 2021, and updated in May 2022, in Medline, Cochrane Library, Cinahl, AMED, and PEDro. Risk of bias and applicability of the included studies were appraised using the Quality Assessment of Diagnostic Accuracy Studies-2 tool and the Quality Appraisal of Reliability Studies tool. Included studies were synthesised narratively. Certainty of evidence was evaluated for each assessment component using GRADE.

### Results

Ten repeated-measures studies were included, involving 193 participants aged 23–62 years. Reported validity of digital physiotherapy assessment ranged from moderate/acceptable to almost perfect/excellent for clinical tests, range of motion, patient-reported outcome

**Data Availability Statement:** All relevant data are within the paper and its Supporting Information files.

**Funding:** The study was partially supported by Research, Education, Development and Innovation primary care, Region Västra Götaland, Sweden. They had no role in the study design, data collection and analysis, decision to publish, or preparation of the manuscript. Open Access funding was provided by the University of Gothenburg Library.

**Competing interests:** The authors have declared that no competing interests exist.

**Abbreviations:** CI, confidence interval; ICC, Interclass correlation coefficient; GRADE, Grading of Recommendations, AssessmentDevelopment and Evaluations; PROM, patient-reported outcome measure; QUADAS-2, Quality Assessment of Diagnostic Accuracy Studies-2 tool; QUAREL, Quality Appraisal Tool for studies of diagnostic Reliability; PRISMA, Preferred Reporting Items for Systematic Reviews and Meta-Analyses; ROM, range of motion; SD, standard deviation; SE, standard error; VAS, visual analogue scale; WHO, World Health Organization; $X_2$, Chi-square.

measures (PROMs), pain, neck posture, and management decisions. Reported validity for assessing spinal posture varied and was for clinical observations unacceptably low. Reported validity and reliability for digital diagnosis ranged from moderate to almost perfect for exact+similar agreement, but was considerably lower when constrained to exact agreement. Reported reliability was excellent for digital assessment of clinical tests, range of motion, pain, neck posture, and PROMs. Certainty of evidence varied from very low to high, with PROMs and pain assessment obtaining the highest certainty. Patients were satisfied with their digital assessment, but did not perceive it as good as face-to-face assessment.

## Discussion

Evidence ranging from very low to high certainty suggests that validity and reliability of digital physiotherapy assessments are acceptable to excellent for several assessment components. Digital physiotherapy assessment may be a viable alternative to face-to-face assessment for patients who are likely to benefit from the accessibility and convenience of remote access.

## Trial registration

The review was registered in the PROSPERO database, CRD42021277624.

## Introduction

Since 2005, the World Health Organization (WHO) has called on nations to work strategically with development and implementation of eHealth, and considerable progress has been made in recent years [1]. The term eHealth has evolved into 'digital health', a broad umbrella term defined as 'the field of knowledge and practice associated with the development and use of digital technologies to improve health' [1]. To further support countries to develop and consolidate national strategies for eHealth and digital health, the WHO has decided on a Global strategy on digital health 2020–2025, envisioning improvements in 'health for everyone, everywhere by accelerating the development and adoption of appropriate, accessible, affordable, scalable and sustainable person-centric digital health solutions. . .' [1].

Digital health encompasses categories such as mobile health, health information technology and telehealth, and covers digital solutions using electronic information and telecommunication technologies to support long-distance clinical health care, patient and professional health-related education, health administration, and public health [2, 3]. It also includes digital consumers, such as patients seeking health care via their computer or smartphone [1]. The COVID-19 pandemic has accelerated the development of digital solutions in health care and brought rapidly increased access to digital health during recent years [4].

Physiotherapy assessment of musculoskeletal disorders is traditionally performed in a face-to-face session at an outpatient clinic, with the patient and the physiotherapist in the same room. The assessment typically comprises several components, including observation, postural examination, gait analysis, joint range of motion testing, palpation, and other clinical examinations and tests, which vary with pain location and suspected diagnosis. The assessment is based on the patient history, and the whole assessment is interpreted by the physiotherapist in order to make a diagnosis and management decisions. Physiotherapy assessment using real-time digital technology typically involves audio and visual communication between the patient and the physiotherapist via a videoconferencing system, using a computer, tablet or smartphone.

Two prior systematic reviews have examined the literature on physiotherapy assessment for musculoskeletal disorders using real-time digital technology, up until May 2015 [5] and December 2016 [6], respectively. Both reviews suggested that some components of physiotherapy assessments via telehealth video solutions could possibly be reliable and valid in musculoskeletal disorders, although both reliability and validity showed high variability [5, 6]. In an evaluation of telehealth physiotherapy in response to COVID-19, patients reported being satisfied and willing to use telehealth physiotherapy again [7]. However, the body of evidence regarding digital physiotherapy assessment AMED for musculoskeletal disorders is still small.

Due to the rapid acceleration of digital solutions during the COVID-19 pandemic, there is a need to systematically review recently published studies in addition to those already included in earlier reviews. Furthermore, an important limitation of previous reviews is the lack of assessment of the certainty in the body of evidence. Therefore, this systematic review aimed to assess the certainty in the evidence for physiotherapy assessment using real-time digital technology compared with conventional face-to-face physiotherapy assessment for patients with musculoskeletal conditions, concerning validity, reliability, feasibility, patient satisfaction, physiotherapy satisfaction, adverse events, clinical management, and cost-effectiveness.

## Methods

This systematic review is reported according to the Preferred Reporting Items for Systematic Reviews and Meta-Analyses (PRISMA) statement [8]. A populated PRISMA checklist can be found as S1 Checklist in S1 File. The review protocol was prospectively registered on PROSPERO on 23 Sept 2021 (CRD277624).

### Eligibility criteria

Articles were eligible for inclusion if they reported original studies of any design in which digital physiotherapy assessment using real-time video technology was compared with conventional, face-to-face physiotherapy assessment for patients with musculoskeletal disorders.

Inclusion criteria: Included articles had to meet the criteria defined in a PICO (population, index test, comparator test, and outcome) (Box 1).

We defined validity as the extent of agreement between digital assessment and face-to-face assessment, regardless of the terminology used in the included articles. We defined reliability as the agreement between two or more observations of the same entity; inter-rater reliability refers to the agreement between two or more raters who observe the same entity, intra-rater reliability refers to the agreement between two or more observations of the same entity by a single rater [9].

No exclusion criteria were applied.

### Information sources

Systematic database searches were performed by a medical librarian in Medline (OvidSP), Cochrane Library, Cinahl (EBSCO), AMED (EBSCO), and PEDro. All searches were done on 26 May 2021. When possible, the searches included conference abstracts, theses, and reports in the above databases, in addition to published articles. Reference lists of relevant papers were scrutinised for additional articles for potential inclusion. Grey literature was also searched via the website of the Swedish Agency for Health Technology Assessment and Assessment of Social Services and Google. For practical reasons, we only screened the first 100 hits of the Google search. An update literature search was performed by one of the reviewers in the same five databases on 25 May 2022. Search strategies for the respective database were copied into the search fields to ensure they were identical.

### Box 1. Inclusion criteria

| | |
|---|---|
| Population: | Patients with musculoskeletal disorders |
| Index test: | Physiotherapy assessment using real-time digital video technology |
| Comparator test/reference standard: | Conventional, face-to-face physiotherapy assessment |
| Outcomes: | *Primary outcomes* |
| | Validity for physiotherapy assessment components |
| | Inter-rater and intra-rater reliability for physiotherapy assessment components |
| | *Secondary outcomes* |
| | Feasibility |
| | Patient satisfaction |
| | Physiotherapist satisfaction |
| | Clinical management |
| | Adverse events |
| | Cost-effectiveness |

## Search strategy

A search strategy was developed for Medline by the authors together with a medical librarian and adapted for the other databases. A combination of key words and subject headings was used to encompass three concepts: musculoskeletal disorders, digital assessment, and physiotherapy. The full search strategy and results for each database are presented in S2 File. No limitations were set for publication period or language.

## Selection process

In the first step of the selection process, reviewers in pairs independently screened titles and abstracts and removed duplicates and records that were clearly not eligible for this systematic review. The Rayyan app [10] was used for the screening process. In a second step, each reviewer pair performed full-text screening independently. Reviewers were blinded to each other's decisions. Any disagreements regarding inclusion decision were resolved by consensus. Records from the update literature search were also screened independently by two reviewers.

## Data collection process

Data from the included studies were extracted into a purpose-built datasheet by one reviewer and checked for accuracy by another and presented in tables. Data extracted included first author, publication year, country of origin, study design, setting, population/musculoskeletal condition (including age, gender), sample size, intervention/index test, comparator/reference standard, assessment procedure, time interval between assessments, outcome measures and main findings. Only data from populations with musculoskeletal conditions were extracted.

## Risk of bias and applicability

Two reviewers independently appraised the quality of each of the included studies using the Quality Assessment of Diagnostic Accuracy Studies (QUADAS)-2 tool for the studies reporting validity [11] and, for the studies reporting reliability, the Quality Appraisal Tool for studies of diagnostic Reliability (QAREL) [12]. The QUADAS-2 comprises 14 items in four domains:

patient selection, index test, reference standard, and flow and timing. Eleven items pertain to risk of bias and three items pertain to concerns regarding applicability of the study findings to the review question. The item "Was a pre-specified threshold used", a so called 'signaling question', was not considered applicable to our review and was removed. Risk of bias was judged as low, high, or unclear for each domain, and a study was judged to be of overall low risk of bias if the answers to the signaling questions in all domains were "yes". The study was judged as having no concerns regarding applicability if all domains were in line with the review question; otherwise it was judged as having concerns [11].

The QAREL comprises 11 items that measure sampling bias/representativeness of subjects and raters, rater blinding, order of examination, time interval among repeated measures, application and interpretation of the assessment, and statistical analysis. Any disagreements in the assessment were discussed among all reviewers and resolved in consensus.

## Data analysis and interpretation

The primary outcomes validity and reliability were analysed separately for each component of the physiotherapy assessment, as well as for clinical management decisions. All reported effect measures were extracted and analysed. To interpret the strength of agreement, we followed published reporting guidelines for reliability studies [13], and categorised agreement values according to established cut-offs (Table 1). For the outcome patient satisfaction, means and medians were calculated.

## Synthesis of results

The included studies are synthesised narratively due to considerable heterogeneity in investigated conditions, reference standard used, and outcomes reported in the included studies. Data were not reported in sufficient detail to be able to perform meta-analyses, i.e., dispersion measures or 95% confidence intervals (CIs) were not generally reported. The heterogeneity restricted possibilities of performing sub-group and sensitivity analyses. Validity and reliability

**Table 1. Cut-offs for strength of agreement.**

| Source | Statistical test | Cut-offs | Interpretation |
|---|---|---|---|
| Landis and Koch 1977 [14] | Kappa | <0.00 | Poor |
| | | 0.00 to 0.20 | Slight |
| | Weighted kappa (Percentage agreement) | 0.21 to 0.40 | Fair |
| | | 0.41 to 0.60 | Moderate |
| | | 0.61 to 0.80 | Substantial |
| | | 0.81 to 1.00 | Almost perfect |
| Krippendorff 2004 [15] | Krippendorff's α | ≥ 0.67 | Acceptable |
| | | ≥ 0.80 | Reliable |
| George and Mallory 2003 [16] | Cronbach's α | < 0.5 | Unacceptable |
| | | > 0.5 | Poor |
| | | > 0.6 | Questionable |
| | | > 0.7 | Acceptable |
| | | > 0.8 | Good |
| | | > 0.9 | Excellent |
| Terwee 2007 [17] | Intraclass correlation coefficient (ICC) | ≥ 0.70 or | Acceptable |
| Shrout and Fleiss 1979 [18] | | 0.0 to 0.40 | Poor |
| | | 0.40 to 0.75 | Fair to moderate |
| | | 0.75 to 1.00 | Excellent |

measures used in the included studies varied, and findings were synthesised according to the cut-offs in Table 1.

## Certainty of evidence assessment

Certainty of evidence across studies for the validity and reliability of the various components of the digital assessment and for clinical management decisions was assessed using the Grading of Recommendations, Assessment, Development and Evaluations (GRADE) approach [19, 20]. The GRADE assessments were applied to the following domains: study limitations, consistency, directness, precision, and reporting bias; and certainty of evidence was categorised in four levels: high, moderate, low, or very low certainty of evidence. Following GRADE guidance, certainty of evidence was rated high for appropriately designed diagnostic studies in patients with diagnostic uncertainty and direct comparisons of test results with an appropriate reference standard, and then rated down if there were concerns in any of the domains [19, 20]. Validity and reliability for all patient-reported outcomes were assessed together. Patho-anatomical and systems diagnoses were assessed together. For patient satisfaction, certainty of evidence was not assessed because no comparative data were available for this assessment component.

## Results

### Study selection

The database searches yielded a total of 1649 records. After duplicates were removed, 1121 records remained. After screening titles and abstracts, 21 studies remained, including one found through citation search. Ten studies were finally included in the review. Excluded studies are listed in S3 File. Study selection and screening results are presented in Fig 1. The update literature search resulted in 215 records but did not identify any new studies that met our PICO.

### Study characteristics

Characteristics of the included studies are presented in Table 2. All ten included studies used a repeated measures design with randomised order of assessment. Seven of the studies were conducted in Australia, by the same research group [21–27]. Several of these studies were conducted in a university-based physiotherapy musculoskeletal and sports injury clinic and all used the eHAB telerehabilitation system (Uniquest/NeoRehab, Brisbane, Queensland, Australia). This is a videoconferencing system that includes a motion analysis tool that facilitates data extraction for a battery of physical tests, including joint range of motion (ROM), linear distance and postural analysis [22]. The other three studies were conducted in Canada [28], Malaysia [29] and Spain [30], and used other advanced telerehabilitation systems with special analysis tools. In all included studies, the reference standard consisted of face-to-face assessment conducted according to usual physiotherapy clinical practice. Both index and reference standard comprised a variety of tests depending on condition and the examiner's discretion.

Three studies [21, 27, 28] investigated validity only, the remaining seven studies [22–26, 29, 30] investigated both validity and reliability of digital assessment versus face-to-face assessment. Validity and reliability were most commonly assessed using percentage of exact or similar agreement. Krippendorff's α, mean difference, Chi-square ($X_2$), Cohen's kappa or weighted kappa, and Gwet's first order agreement coefficient were also used for binary and categorical data. For continuous data, Bland-Altman's limit of agreement, standard error of measurement, coefficient of variation, minimal detectable change, interclass correlation coefficient (ICC), Cronbach's α, and Pearson's correlation were used.

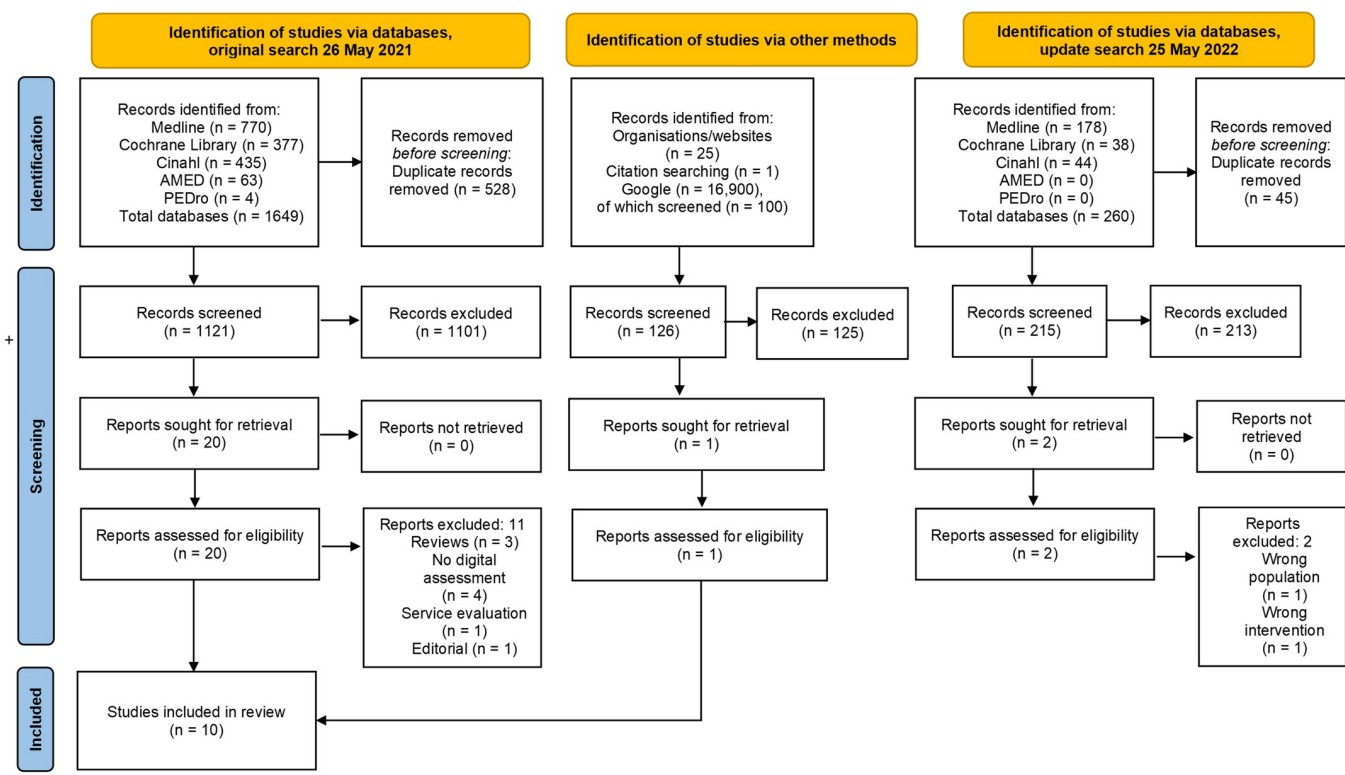

From: Page MJ, McKenzie JE, Bossuyt PM, Boutron I, Hoffmann TC, Mulrow CD, et al. The PRISMA 2020 statement: an updated guideline for reporting systematic reviews. BMJ 2021;372:n71.
doi: 10.1136/bmj.n71. For more information, visit: http://www.prisma-statement.org/

**Fig 1. Flow diagram of literature searches and search results.**

Patient satisfaction was investigated in seven studies [21–27]. Feasibility, physiotherapist satisfaction, adverse events, and cost-effectiveness were not reported in any of the included studies.

Four of the studies were conducted amongst patients with lower limb disorders [23–25], two amongst patients with upper limb disorders [23–25, 28], and three amongst patients with spinal pain [22, 26]. One study [27, 29, 30] was conducted in a mix of patients with lower limb, upper limb and spinal problems. Most studies recruited participants amongst patients who sought help at a physiotherapy clinic. One study [28] recruited patients from a university hospital prior to being discharged after knee arthroplasty, and one study [27] recruited participants through advertising in local media.

A total of 193 patients participated in the studies, ranging from 10 to 42 patients in each study. The mean age of the participants varied from 23 to 62 years. The proportion of women ranged between 10% and 74%. No studies were found which investigated children or older adults.

## Risk of bias and applicability

All included studies reported validity and were assessed using QUADAS-2. Seven of the studies were assessed as being of high quality with low risk of bias and three as having a high risk of bias, mainly due to problems related to patient selection (Table 3). All but one of the studies were assessed as having low concern regarding applicability to our review question. One study [27] included participants with a history of, but not necessarily current, low back pain which reduced the applicability to our research question.

**Table 2. Characteristics of included studies.**

| First author, year (REF) | Country | Study design | Setting | Patient population (condition) | Total n (% women) | Patient population mean age years (SD) | Index test | Reference standard | Time interval between tests | Patient outcomes | Study outcomes | Outcome measures |
|---|---|---|---|---|---|---|---|---|---|---|---|---|
| Cabana 2010 [28] | Canada | Repeated measures design with alternate order of assessment and varied settings for the intervention | Canadian university hospital/ageing research centre/patients' homes | Patients recently discharged after total knee arthroplasty | 15 (47%) | 62 (SD NR) | Digital assessment performed by a remote physiotherapist using a commercial telerehabilitation system (Tandberg 500MXP) | Face-to face physical examination performed by a physiotherapist | 1 day | Physical examination findings: knee flexion/extension ROM*; scar condition; knee joint swelling; 30 second chair stand test; timed up and go; gait (Tinetti) and balance (Berg balance test) | Validity | Bland-Altman's limit of agreement; Krippendorffs alpha; mean difference |
| Cottrell 2018 [21] | Australia | Repeated measures design with block-randomised order of both assessment and physiotherapist | Australian advanced-practice physiotherapy hospital-based screening clinic | Adult patients with chronic, non-urgent lumbar, knee or shoulder pain | 42 (57%) | 52.7 (14.5) | Digital assessment performed by a remote advanced-practice physiotherapist using a commercial telerehabilitation system (eHAB) | Face-to-face physical examination performed by an advanced-practice physiotherapist | Single session (30 minutes between assessments) | None assessed | Validity regarding clinical management decisions: recommended management pathways, referral to allied health professions, primary clinical diagnosis; participant satisfaction | Percentage exact or similar agreement; proportion specific agreement; Cohen's kappa; Gwet's first order agreement coefficient; descriptive statistics |
| Lade 2012 [22] | Australia | Repeated measures design with randomised order of assessment | Australian physiotherapy musculoskeletal and sports injury clinic | Adult patients with elbow pain or dysfunction | 10 (10%) | 38 (13) | Digital assessment performed by a remote final year physiotherapy honours students using a commercial telerehabilitation system (eHAB) | Face-to-face physical examination by final year physiotherapy honours students | Single session (time interval not specified); intra-rater reassessment 6 weeks | Patho-anatomical diagnoses, systems diagnosis, physical examination findings: ROM*, nerve test, orthopaedic tests, pain response, joint assessment, strength, limiting factor | Validity, inter- and intra-rater reliability, participant satisfaction | Percentage exact or similar agreement; quadratically weighted kappa and descriptive statistics |
| Mani 2021 [29] | Malaysia | Repeated measures design with randomised order of assessment | Malaysian university physiotherapy lab | Adult patients (18–55) with non-specific neck pain | 11 (60%) | 32.7 (10.9), range 18–50 | Digital assessment performed by a remote physiotherapist using a university-developed telerehabilitation system (telePTsys) | Face-to-face physical examination performed by a physiotherapist | Single session (30 minutes between assessments); intra-rater reassessment 1 month | Pain, disability, physical examination findings: posture, AROM*, DNF* endurance | Validity, inter- and intra-rater reliability | Percentage agreement; chi-squared statistics; Bland-Altman's limit of agreement; standard error of measurement; coefficient of variation; minimal detectable change; interclass correlation coefficient |

(*Continued*)

**Table 2.** (Continued)

| First author, year (REF) | Country | Study design | Setting | Patient population (condition) | Total n (% women) | Patient population mean age years (SD) | Index test | Reference standard | Time interval between tests | Patient outcomes | Study outcomes | Outcome measures |
|---|---|---|---|---|---|---|---|---|---|---|---|---|
| Palacin-Marin 2013 [30] | Spain | Repeated measures design with randomised order of assessment | Spanish primary care centre | Adult patients with chronic low back pain | 15 (60%) | 37 (SD NR) | Digital assessment performed by a remote physiotherapist using a university/healthcare telerehabilitation system (TPLUFIB-WEB) | Face-to-face physical intervention performed by a physiotherapist | Single session (30 minutes between assessments); intra-rater reassessment 1 month | Pain, disability, quality of life, kinesiophobia, physical examination findings: lumbar spine mobility, back muscle endurance, lumbar motor control | Validity, inter- and intra-rater reliability | Bland-Altman's limit of agreement; Cronbach alpha; interclass correlation coefficient |
| Richardson 2017 [23] | Australia | Repeated measures design with randomised order of assessment | Australian physiotherapy musculoskeletal and sports injury clinic | Adult patients with knee pain | 18 (56%) | 23 (7) | Digital assessment performed by a remote physiotherapist using a commercial telerehabilitation system (eHAB) | Face-to-face physical examination performed by a physiotherapist | Single session (10 minutes between assessments); intra-rater reassessment 1 month | Patho-anatomical diagnoses, systems diagnosis, physical examination findings: ROM*, pain response, pain rating, walking, biomechanical issues, MCL test* | Validity, inter- and intra-rater reliability, participant satisfaction | Percentage exact or similar agreement, chi-squared, quadratically weighted kappa and descriptive statistics |
| Russel 2010a [24] | Australia | Repeated measures design with randomised order of assessment | Australian physiotherapy musculoskeletal and sports injury clinic | Adult patients with ankle pain or dysfunction | 15 (67%) | 24.5 (10.8) | Digital assessment performed by remote final year physiotherapy honours students using a commercial telerehabilitation system (eHAB) | Face-to-face physical examination performed by final year physiotherapy honours students | Single session (10 minutes between assessments); intra-rater reassessment 1 month | Patho-anatomical diagnoses, systems diagnosis, physical examination findings: ROM*, CFL test*, gait analysis, single leg squat, pain response and pain rating | Validity, inter- and intra-rater reliability, participant satisfaction | Percentage exact agreement; chi-squared, quadratically weighted kappa and descriptive statistics |
| Russel 2010b [25] | Australia | Repeated measures design with randomised order of assessment | Australian physiotherapy musculoskeletal and sports injury clinic | Adult patients with lower limb pain self-evaluated as non-articular | 19 (74%) | 26 (13) | Digital assessment performed by a remote physiotherapist using a commercial telerehabilitation system (eHAB) | Face-to-face physical examination performed by a physiotherapist | Single session (10 minutes between assessments); intra-rater reassessment 1 month | Patho-anatomical diagnoses, systems diagnosis. | Validity, inter- and intra-rater reliability, participant satisfaction | Percentage exact or similar agreement; chi-squared, quadratically weighted kappa and descriptive statistics |
| Steele 2012 [26] | Australia | Repeated measures design with block-randomised order of assessment | Australian university physiotherapy department | Adult participants with shoulder pain | 22 (27%) | 30.7 (14.2), range 18-60 | Digital assessment performed by remote final year physiotherapy honours students using a commercial telerehabilitation system (eHAB) | Face-to-face physical examination by final year physiotherapy honours students | Single session (within 1.5 hours); intra-rater reassessment 6 weeks | Pathoanatomical diagnosis, systems diagnosis, physical examination findings: postural analysis, joint assessment, ROM*, pain response, strength, special orthopaedic tests, neural testing | Validity, inter-rater reliability, intra-rater reliability, participant satisfaction | Percentage exact or similar agreement; chi-squared and descriptive statistics |

(Continued)

**Table 2.** (Continued)

| First author, year (REF) | Country | Study design | Setting | Patient population (condition) | Total n (% women) | Patient population mean age years (SD) | Index test | Reference standard | Time interval between tests | Patient outcomes | Study outcomes | Outcome measures |
|---|---|---|---|---|---|---|---|---|---|---|---|---|
| Truter 2014 [27] | Australia | Repeated measures design with block-randomised order of assessment | Australian small town hospital | Adults in small town community with non-urgent current or recent low back pain | 26 (58%) | 43 (SD NR) | Digital assessment performed by a remote physiotherapist using a commercial telerehabilitation system (eHAB) | Face-to-face physical examination performed by a physiotherapist | Single session (5–10 min between assessments) | Physical examination findings: pain, posture, AROM*, SLR test* | Validity, participant satisfacion. | Percentage exact or similar agreement; Pearson's correlation; quadratically weighted kappa and descriptive statistics |

AROM: active range of motion; CFL: calcaneo-fibular ligament; DNF: deep neck flexor; NR: not reported; ROM: range of motion; SD: standard deviation; SLR: straight leg raise

Adult ≥18

**Table 3. QUADAS-2 risk of bias assessment of included studies reporting validity.**

| Study | Risk of bias | | | | Applicability concerns | | |
|---|---|---|---|---|---|---|---|
| | Patient selection | Index test | Reference standard | Flow and timing | Patient selection | Index test | Reference standard |
| Cabana 2010 [28] | ☹ | ☺ | ☺ | ☺ | ☺ | ☺ | ☺ |
| Cottrell 2018 [21] | ☹ | ☺ | ☺ | ☺ | ☺ | ☺ | ☺ |
| Lade 2012 [22] | ☺ | ☺ | ☺ | ☺ | ☺ | ☺ | ☺ |
| Mani 2021 [29] | ☹ | ☺ | ☺ | ☺ | ☺ | ☺ | ☺ |
| Palacín-Marín 2013 [30] | ☺ | ☺ | ☺ | ☺ | ☺ | ☺ | ☺ |
| Richardson 2017 [23] | ☺ | ☺ | ☺ | ☺ | ☺ | ☺ | ☺ |
| Russel 2010a [24] | ☺ | ☺ | ☺ | ☺ | ☺ | ☺ | ☺ |
| Russel 2010b [25] | ☺ | ☺ | ☺ | ☺ | ☺ | ☺ | ☺ |
| Steele 2012 [26] | ☺ | ☺ | ☺ | ☺ | ☺ | ☺ | ☺ |
| Truter 2014 [27] | ☺ | ☺ | ☺ | ☺ | ☹ | ☺ | ☺ |

☺ Low risk ☹ High risk

Risk of bias and concerns regarding applicability across the included studies are summarised in Fig 2.

Seven of the included studies reported reliability and were also assessed using the QAREL checklist (Table 4). Total score ranged between 8 and 10, out of a maximum possible score of 11, with most concerns relating to raters not being blinded to their own prior findings (item 4) and raters not being representative of those to whom the authors intended the results to be applied (item 2). In three of the studies [22, 24, 26] the raters were physiotherapy students, which reduces the transferability of the findings to regular physiotherapy practice.

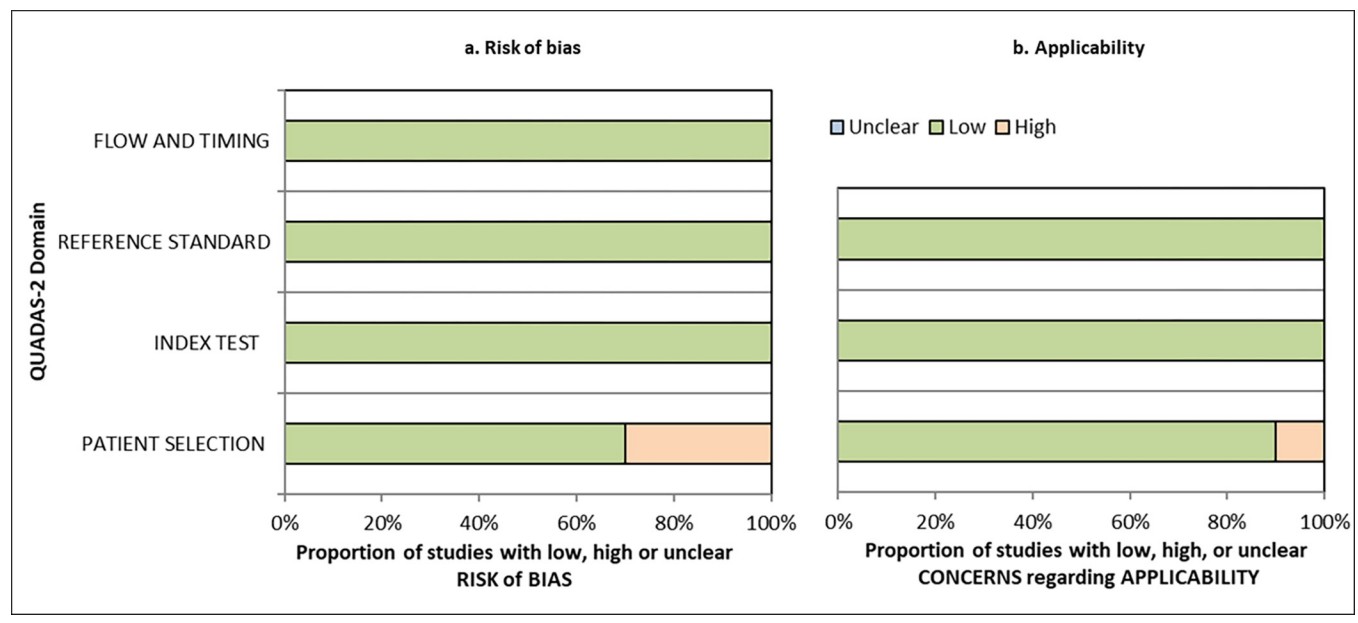

**Fig 2. Proportion of studies assessed as low, high, or unclear risk of bias and applicability.**

**Table 4. QAREL risk of bias assessment of included studies reporting reliability.**

| Study | QAREL items | | | | | | | | | | | Total |
|---|---|---|---|---|---|---|---|---|---|---|---|---|
| | 1. Subjects representative? | 2. Raters representative? | 3. Raters blinded to other raters'findings? | 4. Raters blinded to own prior findings? | 5. Raters blinded to reference standard results? | 6. Raters blinded to clinical information not part of the test? | 7. Raters blinded to additional cues? | 8. Order varied? | 9. Stability taken into account/ suitability of timeinterval? | 10. Test applied correctly and interpreted appropriately? | 11. Appropriate statistical measures? | |
| Lade 2012 [22] | Y | N | Y | N | Y | Y | Y | Y | Y | Y | Y | **9/11** |
| Mani 2021 [29] | Y | Y | Y | N | Y | Y | Y | Y | Y | Y | Y | **10/11** |
| Palacín-Marín 2013 [30] | Y | Y | Y | N | Y | Y | Y | Y | Y | Y | Y | **10/11** |
| Richardson 2017 [23] | Y | Y | Y | N | Y | Y | Y | Y | Y | Y | Y | **10/11** |
| Russell 2010a [24] | Y | N | Y | N | Y | Y | Y | Y | Y | Y | Y | **9/11** |
| Russell 2010b [25] | Y | Y | Y | N | Y | Y | Y | Y | Y | Y | Y | **10/11** |
| Steele 2012 [26] | N | N | Y | N | Y | Y | Y | Y | Y | Y | Y | **8/11** |

Y:Yes; N:No

## Primary outcomes

Certainty of evidence for the outcomes validity and reliability are presented below for each component of the digital assessment and summarised in Table 5. All extracted data for components investigated in more than one study are presented in S4 File.

**Clinical tests.** Clinical tests included orthopaedic and functional tests (except ROM which is described separately) and were assessed in nine studies, with a total of 151 patients [22–30]. A large variety of tests were used for clinical assessments, depending on the patient populations in the included studies, as well as the individual patient history and suspected diagnosis (S4 File). In three of the studies [23–25], findings from clinical examinations were only analysed and presented in groups of either binary or categorical data, and not specified for specific tests.

Validity of digitally performed clinical tests was calculated using different measures. Six studies [22–27] reported varying validity, from moderate to almost perfect agreement for various strength, balance, nerve and other orthopaedic and functional tests (percentage agreement 46% to 99%). Lowest agreement was reported for upper limb nerve tests and joint assessment. One study [28] reported percentage differences in various knee tests between -6% and 5% (95% CI -33% to 29%), and Krippendorff's α between 0.76 and 0.87, corresponding to acceptable or reliable agreement. One study [30] reported Cronbach's α for functional and strength tests ranging between 0.80 and 0.97, corresponding to good to excellent agreement. One study [29] reported a significant mean difference of -2.28 (-4.46 to -0.11) seconds, Bland-Altman's limits of agreement -8.25 to 3.68 for a neck endurance test. Three studies [23–25] reported kappa values between 0.64 and 0.92 for mixed lower limb tests and one study [27] reported a kappa of 0.64 for the straight leg raise test for low back pain; all corresponding to substantial to almost perfect agreement. Five studies [23–27] reported $\chi 2$ values between 0.76 and 400.4 for various clinical tests for shoulder, lower limb and back disorders. Certainty of evidence for the validity of digitally performed clinical tests was assessed as very low, due to some study limitations, serious inconsistency, serious indirectness, and very serious imprecision.

**Table 5. Summary of findings and GRADE assessment.**

| Assessment component | Population | No. of studies (patients) | Validity (% agreement, unless otherwise stated) | Certainty of evidence (GRADE) | No. of studies (patients) | Inter-rater reliability (% agreement, unless otherwise stated) | Intra-rater reliability (% agreement, unless otherwise stated) | Certainty of evidence (GRADE) |
|---|---|---|---|---|---|---|---|---|
| Clinical tests (including orthopaedic and functional tests) | Adult patients with upper limb pain, lower limb pain, neck pain, LBP | 9 (151) | Upper limb: 46% to 90% MODERATE AGREEMENT+ Lower limb: 83% to 99%; Krippendorff's α 0.76 to 0.87; weighted kappa 0.64 to 0.92 ACCEPTABLE AGREEMENT+ Neck: MD -2.3 (95% CI -4.5 to -0.1) seconds Back: 82% to 90%; Cronbach's α 0.80 to 0.97 GOOD AGREEMENT+ | Very low[1] | 7 (110) | Upper limb: 67% to 97% SUBSTANTIAL AGREEMENT+ Lower limb: 93% to 100%; weighted kappa 0.94 to 0.98 Neck: ICC 0.99 (95% CI 0.98 to 0.99) LBP: ICC 0.92 to 0.93 (95% CI 0.91 to 0.94) ALMOST PERFECT/ EXCELLENT AGREEMENT | Upper limb: 81% to 98% Lower limb: 97% to 100%; weighted kappa 0.98 to 0.99 Neck: ICC 0.99 (95% CI 0.98 to 0.99) LBP: ICC 0.94 to 0.95 (95% CI 0.93 to 0.96); Cronbach's α 0.95 ALMOST PERFECT/ EXCELLENT AGREEMENT | Very low[2] |
| Range of motion (ROM) | Adult patients with knee pain, elbow pain, neck pain, LBP, ankle pain, shoulder pain | 8 (132) | Knee, elbow, shoulder, back: 81% to 96% ALMOST PERFECT AGREEMENT Neck (active ROM): MD -1.0 to +1.2 cm (ns) Back: Cronbach's α 0.75 to 0.99 ACCEPTABLE AGREEMENT+ | Moderate[3] | 6 (91) | Elbow, shoulder: 92%-93% Neck, LBP: ICC 0.92 to -0.98 (95% CI 0.90 to 0.99) ALMOST PERFECT/ EXCELLENT AGREEMENT | Elbow, shoulder: 95%-96% Neck, LBP: ICC 0.94 to 0.99 (95% CI 0.89 to 0.99) ALMOST PERFECT/ EXCELLENT AGREEMENT | Moderate[3] |
| Clinical observations | Adult patients with knee arthroplasty | 1 (15) | Krippendorff's α 0.34 UNACCEPTABLE AGREEMENT | Very low[4] | - | NR | NR | |
| PROMs, incl. pain assessed using VAS | Adult patients with neck pain or LBP | 2 (26) | MD -0.68 to 1.06 (95% CI -4.49 to 6.62) Cronbach's α > 0.94 EXCELLENT AGREEMENT | High | 1 (11) | ICC 0.99 to 1.00 (95% CI 0.97 to 1.00) EXCELLENT AGREEMENT | ICC 0.99 (95% CI 0.97 to 1.00) EXCELLENT AGREEMENT | High |
| Pain | Adult patients with elbow pain or shoulder pain | 2 (32) | 77% to 82% SUBSTANTIAL AGREEMENT+ | High | 2 (32) | 97% to 98% ALMOST PERFECT AGREEMENT | 97% ALMOST PERFECT AGREEMENT | Moderate[5] |
| Posture | Adult patients with neck pain or LBP | 2 (37) | Neck: MD -0.96° to -0.32° (95% CI -1.49 to 0.25) Back: 25% to 75%; Kappa -0.20 to 0.19 POOR AGREEMENT+ | Very low[6] | 1 (11) | Neck: ICC 0.93 to 0.99 (95% CI 0.70 to 0.99) EXCELLENT AGREEMENT | ICC 0.93 to 1.00 (95% CI 0.69 to 0.99) EXCELLENT AGREEMENT | Low[7] |
| Diagnosis | Adult patients with lumbar, knee, ankle, shoulder and elbow pain | 6 (126) | Patho-anatomical diagnosis Exact: 18% to 68% Similar: 11% to 50% Exact+similar: 59% to 93% MODERATE AGREEMENT+ Systems diagnosis 73% to 94% SUBSTANTIAL AGREEMENT+ | Low[8] | | Patho-anatomical diagnosis Exact: 18% to 67% Similar: 26% to 55% Exact+similar: 73% to 100% SUBSTANTIAL AGREEMENT+ Systems diagnosis 64% to 93% SUBSTANTIAL AGREEMENT+ | Patho-anatomical diagnosis Exact: 41% to 93% Similar: 7% to 59% Exact+similar: 82% to 100% ALMOST PERFECT AGREEMENT Systems diagnosis 82% to 100% ALMOST PERFECT | Low[8] |

*(Continued)*

**Table 5.** (Continued)

| Assessment component | Population | No. of studies (patients) | Validity (% agreement, unless otherwise stated) | Certainty of evidence (GRADE) | No. of studies (patients) | Inter-rater reliability (% agreement, unless otherwise stated) | Intra-rater reliability (% agreement, unless otherwise stated) | Certainty of evidence (GRADE) |
|---|---|---|---|---|---|---|---|---|
| Clinical manage-ment | Adult patients with lumbar, knee or shoulder pain | 1 (42) | 83%<br>Gwet's AC1: 0.83<br>ALMOST PERFECT AGREEMENT | Low[9] | - | NR | NR | |

CI: Confidence interval; LBP: Low back pain; ICC: Intraclass correlations; MD: Mean difference; NR: Not reported; ns: not significant. PA: Percentage agreement; PROMs: Patient-reported outcome measures: ROM: Range of motion.

Landis och Koch's cut-offs from poor to almost perfect are used to describe PA and weighted kappa; Krippendorff's cut-offs from unacceptable to reliable for Krippendorff's α; George and Mallery's cut-offs from unacceptable to excellent for Cronbach's α; and Shrout and Fleiss' cut-offs from poor to excellent for ICC. The lowest cut-off value is shown in the table with a + indicating when there were also higher values.

[1]Downgraded one level due to serious inconsistency (different tests used, unclear methods, and heterogeneity in agreement), one level due to serious indirectness (some concerns regarding applicability/directness, patient selection, applicability of digital system and methods used), and one level due to very serious imprecision (wide or missing CIs).

[2]Downgraded one level due to some study limitations (raters not representative, raters not blinded to own prior findings) and some inconsistency (large variation in PA among different tests), one level due to serious indirectness (concerns about the applicability of digital system used, unclear tests) and one level due to very serious imprecision (wide or missing CIs for many of the tests).

[3]Downgraded one level due to some study limitations (concerns about patient selection), and serious indirectness (concerns about patient selection and applicability of digital system used in other countries/contexts, for example special technical programs, e-goniometer)

[4]Downgraded one level due to study limitations (unclear patient selection, no consecutive or random sampling) and serious indirectness (only one study, one condition), and two levels due to very serious imprecision (one small study, very wide CIs)

[5]Downgraded one level due to some study limitations (raters not blinded to own prior findings (intra-rater reliability), some uncertainty regarding directness (student raters not representative, few women

[6]Downgraded one level due to some study limitations (no consecutive or random sampling) and uncertain precision (CIs not reported in one of the studies), one level due to serious inconsistency (variation in validity between studies and large variation in PA in one of the studies), and one level due to serious indirectness (concerns regarding applicability of digital systems and neck measurement method used)

[7]Downgraded one level due to some study limitations (no consecutive or random sampling, raters not blinded for intra-rater assessment) and uncertain precision (one small study), and one level due to serious indirectness (applicability of digital system and neck measurement method used, patient selection)

[8]Downgraded one level due to some study limitations (no consecutive or random sampling, raters not blinded for intra-rater assessment) and some concerns regarding directness (applicability of digital systems used, raters not representative), and one level due to serious inconsistency (large variation in PA) and uncertain precision (CIs rarely reported)

[9]Downgraded one level due to serious study limitations (not a consecutive or random sampling) and one level due to uncertainty regarding directness (applicability of digital system used) and precision (only one study).

Five studies [22–26] reported substantial to almost perfect percentage agreement for inter-rater reliability of upper and lower limb tests ranging from 67% to 100%. Two studies [29, 30] reported excellent ICC values of 0.92 to 0.99 for neck and back endurance and functional tests. For back endurance and functional tests, excellent Cronbach α of 0.93 was reported [30]. Three studies [23–25] reported almost perfect weighted kappas for lower limb tests, ranging from 0.94 to 0.98. Four studies [23–26] reported $\chi 2$ values between 7.2 and 1549.90 for upper and lower limb tests. Intra-rater reliability for upper and lower limb tests was assessed in five of the included studies [22–26] as almost perfect, with percentage agreement ranging from 81% to 100%. Two studies [29, 30] reported excellent ICC, 0.94 to 0.99 for neck and back endurance and functional tests, and one [30] excellent Cronbach α of 0.95 for back endurance and functional tests. Three studies [23–25] reported almost perfect weighted kappas of 0.98 to 0.99 for lower limb tests. Four studies [23–26] reported $\chi 2$ values between 51.00 and 1795.95 for upper and lower limb tests. Certainty of evidence for reliability of digitally performed

clinical tests was assessed as very low, due to some study limitations and some inconsistency, serious indirectness, and very serious imprecision.

**Range of motion.** Range of motion was assessed in eight studies, with a total of 132 patients [22–24, 26–30]. However, two studies [23, 24] did not report specific results for ROM.

For validity, three studies [22, 26, 27] reported almost perfect agreement between digital and face-to-face assessment for elbow, shoulder and back disorders, with percentage agreement ranging from 81% to 96% (S4 File). Three studies [28–30] reported small mean and percentage differences for knee, neck and back disorders, of which one [28] reported reliable agreement with Krippendorff's α values ≥ 0.80 and one [30] reported acceptable to excellent agreement with Cronbach's α ranging from 0.75 to 0.99.

Four studies [22, 26, 29, 30] reported almost perfect inter-rater reliability, with percentage agreement of 92–93% for elbow and shoulder disorders [22, 26] and excellent agreement (ICC ≥ 0.92) for neck and back disorders [29, 30]. For intra-rater reliability, the same four studies reported almost perfect percentage agreement of 95–96% for elbow and shoulder disorders [22, 26], and excellent agreement (ICC ≥ 0.94) for neck and back disorders [29, 30].

Certainty of evidence for both validity and reliability of digital ROM assessment was assessed as moderate, with some study limitations and serious indirectness.

**Clinical observations.** Clinical observations were assessed in one study with 15 patients [28]. Specifically, scar observation after knee surgery was investigated, in which coloration, deformity, texture, and contour were rated on Likert scales. Validity of clinical observations was presented as mean difference and Krippendorff's alpha reliability estimate for agreement between methods. Mean difference between methods was -15% (95% CI -85% to +55%), Krippendorff's α 0.34, which indicates unacceptable agreement. Certainty of evidence was assessed as very low, due to study limitations, serious indirectness, and very serious imprecision. Reliability of digital assessment of clinical observations was not investigated.

**Patient-reported outcome measures and pain.** Validity and reliability of patient reported outcome measures (PROMs) and pain were reported in four studies, with a total of 58 patients [22, 26, 29, 30] (S4 File). Disability was reported in two studies [29, 30], measured with the Neck Pain Questionnaire (score 0–36) [31] and the Oswestry Disability Index (ODI) (score 0–50) [32], respectively. Kinesiophobia was evaluated in one study [30] using the Tampa Scale of Kinesiophobia (score 11–44) [33]. Health-related quality of life was evaluated in one study [30] using the Short Form Health Survey questionnaire (score 0–100) [34]. In two studies [29, 30], pain was measured as a PROM using a visual analogue scale (VAS), ranging from 0–100. In two studies [22, 26], pain levels during physical examination performed by the patient themselves were measured either as no change/increased [22, 26] or on a categorical scale 0–10 [26] and compared with pain levels during physical examination performed face-to-face by a physiotherapist.

The validity for digital assessment of pain during a physical examination was moderate with weighted kappa 0.50 [26] (Table c in S4 File). Percentage agreement ranged from 77% to 82% [22, 26]. For PROMs, there were small mean differences (-0.68 to 1.06; 95% CI -4.49 to 6.62) and excellent agreement with Cronbach's α (>0.94) [29, 30]. The greatest discrepancies were for measuring pain as a PROM [29, 30]. Certainty of evidence was assessed as high for the validity of digital pain assessment during physical examination and for digital assessment of PROMs.

Inter- and intra-rater reliability for pain and disability was assessed in three studies [22, 26, 29]. Two studies examined pain during the examination [22, 26] and one [29] pain and disability as PROMs. Inter-rater reliability for same or similar pain assessment was presented as percentage agreement ranging from 97–98% in two studies [22, 26]. The weighted kappa of 0.95 in one study [26] and the ICC of 0.99 in another [29] both indicate excellent levels of

agreement. Inter-rater reliability of disability as a PROM was assessed in one study [29] and was found to be excellent; ICC 1.00. Reported intra-rater reliability for pain assessment was excellent (weighted kappa 0.97 [26], ICC 0.99 [29]). Intra-rater reliability of disability as a PROM was assessed in one study [29] and was found to be excellent; ICC 0.99. Certainty of evidence was assessed as high for reliability of digital assessment of PROMs and as moderate for digital assessment of pain during physical examination, due to some study limitations and indirectness.

**Posture.** Assessment of posture was performed in two studies involving 37 patients [27, 29] (S4 File). Neck posture was measured as sagittal head tilt angle, cranio-cervical angle, and shoulder angle using an e-Goniometer tool [29]. Standing spinal posture in patients with low back pain was measured in the coronal and sagittal planes and included thoracic, lumbar and pelvic symmetry, kyphosis, lordosis and tilt [27]. Reported validity of digital neck posture assessment was high, with mean difference ranging from -0.32˚ to -0.96˚ [29]. Inter- and intra-reliability of digital neck posture assessment were excellent, ICC 0.93–0.99 and ICC 0.93–1.00, respectively [29]. For spinal posture, reported validity of the digital assessment ranged from fair to substantial agreement; percentage exact agreement 25% to 75%, with the lowest agreement being for lumbar lordosis [27]. Reliability for spinal posture assessment was not reported. Certainty of evidence for the validity of digital posture assessment was very low and for reliability low.

**Diagnosis.** Patho-anatomical diagnosis was assessed in six studies [21–26], with a total of 126 patients (S4 File). Systems diagnosis, which describes the anatomical system involved such as muscles, joints or nerves, was assessed in five studies [22–26].

Validity of patho-anatomical diagnosis was assessed as percentage agreement with several studies specifying exact and similar agreement between digital and face-to-face assessments. Percentage exact+similar agreement ranged from 59–93%, which is moderate to almost perfect. Validity of systems diagnosis was assessed in five of the studies with substantial to almost perfect exact percentage agreement ranging between 73% and 94%.

Inter-rater reliability of both patho-anatomical and systems diagnosis was assessed in five studies [22–26] as substantial to almost perfect. Percentage exact+similar agreement for patho-anatomical diagnosis ranged from 73% to 100%. Percentage exact agreement for systems diagnosis ranged from 64% to 93%. Intra-rater reliability of patho-anatomical and systems diagnosis was assessed in five studies [22–26] as almost perfect. Percentage exact+similar agreement for patho-anatomical and exact for systems diagnosis both ranged from 82 to 100%.

Certainty of evidence for both validity and reliability of the diagnosis assessment was rated as low, due to some study limitations, inconsistent results, indirectness and imprecision.

## Secondary outcomes

**Clinical management decisions/management pathway.** Validity of clinical management decisions was investigated in one study with 42 patients [21]. After assessment, examiners chose one of six predefined management pathways. Validity of digital management pathway decisions was almost perfect, exact agreement 83%; Gwet's first order agreement coefficient (AC1) 0.83; SE 0.06; 95% CI 0.70 to 0.95. Reliability of clinical management was not investigated. Certainty of evidence for the validity of digital clinical management decisions was assessed as low, due to serious study limitations and concerns regarding directness and precision.

**Patient satisfaction.** Patient satisfaction with digital assessment was reported in seven studies [21–27]. Overall patient satisfaction with digital assessment was assessed using a VAS (0–100 mm) in all seven studies and presented as mean values ranging from 68 to 89 mm

**Table 6. Mean values of reported patient satisfaction (VAS 0–100).**

| | (1) Confidence with physical self-examination* | (2) Recommend to a friend unable to travel | (3) As good as face-to-face | (4) Visual clarity | (5) Audio clarity | (6) Overall satisfaction |
|---|---|---|---|---|---|---|
| Cottrell 2018 [21] | | | 72 | | | 89 |
| Lade 2012 [22] | 70 | | | | | |
| Richardson 2017 [23] | 65[1] | 75 | 21 | 78 | 78 | 72 |
| Russell 2010a [24] | 63[2] | 80 | 19 | 72 | 78 | 74 |
| Russell 2010b [25] | 73 | 75 | 25 | 80 | 68 | 78 |
| Steele 2012 [26] | 62[3] | 69 | 31 | 70 | 78 | 68 |
| Truter 2014 [27] | 70 | 87 | 30 | 72 | 72 | 85 |
| **Mean** | 67 | 77 | 33 | 74 | 76 | 77 |
| **Median** | 65 | 75 | 27,5 | 72 | 78 | 74 |

*Alternative formulation: [1]How confident were they with the Internet method of a musculoskeletal assessment?; [2]confidence in online assessment?; [3]How beneficial were the Internet examinations?

(mean 77) (Table 6). Patient satisfaction was measured after both the digital and face-to-face assessments and includes subjective comparisons between the two methods. In six of the studies [21, 23–27], participants were asked to rate whether the assessment was considered as good as a face-to-face assessment on a VAS. The responses ranged from 19 to 72 mm (mean 33). Another four questions were rated on a VAS in five studies [23–27]: confidence with physical self-examination (mean 67, range 62–73); recommend to a friend unable to travel (mean 77, range 69–87); visual clarity (mean 74, range 70–80); and audio clarity (mean 76, range 68–82).

One of the studies [21] was at risk of bias due to convenience sampling, and one [27] had concerns regarding applicability due to patient selection.

**Feasibility, physiotherapist satisfaction, adverse events, cost-effectiveness.** None of the studies had investigated any of these outcomes.

## Discussion

This systematic review of digital physiotherapy assessment for musculoskeletal disorders showed that both validity and reliability for different components of the assessment varied, and that certainty of evidence ranged from very low to high depending on the variable. Our review identified relatively few studies, of which three were published since previous reviews, that have examined the validity and reliability of real-time digital physiotherapy assessment of musculoskeletal disorders and clinical management of and patient satisfaction with such assessments. No studies investigating other outcomes of interest were found.

Validity of digital physiotherapy assessment ranged from moderate/acceptable to almost perfect/excellent for several of the different investigated components (clinical tests, ROM, PROMs, pain assessment, management decisions). Validity for digital clinical observations was unacceptably low. Validity and reliability for digital diagnosis ranged from moderate to almost perfect for exact+similar agreement, but with considerably worse results when

constrained to exact agreement between digital and face-to-face examinations. Reliability was excellent for clinical tests, ROM, PROMs and pain assessment. Certainty of evidence varied from very low to high for different components of the digital assessment, with PROMs and pain assessment obtaining the highest certainty. Although patients were satisfied with their digital assessment, it was not perceived as being as good as face-to-face assessment.

Our findings show that several of the clinical tests commonly used in physiotherapy, such as muscle strength tests, orthopaedic tests, nerve tests, and joint assessment, seem particularly difficult to perform using digital technology. In several of the studies, patients were instructed to apply self-resistance and perform muscle tests as well as pain assessment on their own, or a family member or caregiver assisted in performing the tests according to instructions from the remote physiotherapist. This was often challenging and validity for several of the investigated tests, particularly upper limb nerve tests, were lower. However, an important consideration in this regard is that the validity and reliability of many tests performed face-to-face also vary considerably [35–39], as does the clinical relevance of this type of test. Often, specificity is acceptable while sensitivity is lower or more variable [37, 39], suggesting that a test may be appropriate for ruling out certain conditions rather than confirming them.

Posture examination seems to be another challenging component of digital assessment, with poor validity and very low certainty of evidence. Postural inspection is traditionally at the core of the physiotherapy examination, done every day in physiotherapy practices across the globe in patients with spinal pain and other conditions. But are we barking up the wrong tree? How valid are face-to-face posture assessments and how relevant are they? The literature is inconsistent and even contradictory, with review authors drawing different conclusions regarding validity and reliability of clinical posture examination and the association with spinal pain [36, 40–42]. And even if there is an association between poor spinal posture and e.g. low back pain, causation has not been established [40]. The difficulties with performing relevant postural assessments do not seem to be alleviated by digital visual examination but certain digital tools may increase at least the reliability of such assessments.

We identified three relevant other reviews to which we could compare our findings. Although with a high degree of overlap with our review, they differed slightly in scope, populations, and conditions. A recent review by Zischke and colleagues [43] of physiotherapy assessments delivered by "telehealth" was not limited to musculoskeletal disorders and included studies of various designs and from other practice areas. Furthermore, the review was not limited to digital assessment but included even telephone assessments. The review supports our findings of variability in both validity and reliability, and the authors concluded that physiotherapy assessment using synchronous forms of telehealth was valid and reliable for "specific assessment types in limited populations". The review by Mani and colleagues [5] focused on musculoskeletal conditions, but four of eleven included studies were conducted with healthy individuals. They concluded that validity was good and reliability was excellent for most assessments, which is a more positive conclusion than is supported by the data in our review. The discrepancy could be due to their inclusion of studies on healthy individuals and our inclusion of more recent studies; we included three studies published after their review. Grona and colleagues' comprehensive review [6] included both physiotherapy assessment and treatment studies. The authors concluded that many of the physiotherapy assessments were valid and reliable, with some exceptions, which is consistent with our findings. They also appraised the studies included in their review to have high risk of bias.

The quality appraisal of the included studies and certainty of evidence assessments in our review raise some doubts about whether digital assessment is a viable alternative to face-to-face assessments at present. More evidence is needed to confirm particularly the more

complicated components of the assessment, such as performing clinical tests and making diagnostic and management decisions.

Horsley and colleagues [44] suggested in their scoping review that physiotherapy videoconferencing services, including both assessment and treatment, may improve access to care and improve quality of life for individuals with limited access to healthcare services. On the other hand, concerns have been raised about access to digital health care [45–47]. Subgroups identified as having higher access to digital health care are younger, with higher education and higher income, living in neighborhoods with high rates of residential high-speed internet availability and mobile portal use. This may lead to displacement effects of more vulnerable groups who are older and have lower socioeconomic status. Further research on the impact of digital assessment within physiotherapy is needed to determine the equity consequences.

Several of the studies in our review reported patient satisfaction with digital assessment, but ratings varied substantially. Although patients rated that communication and overall expectations were met through digital assessment, many did not rate digital assessment as being as good as face-to face consultations [21–23, 26, 27]. A review of patient and caregiver satisfaction with "telehealth videoconferencing" showed high levels of satisfaction from both perspectives; especially amongst patients living in rural and remote areas in view of the improved access and avoidance of travel afforded by using digital solutions [48]. A recent study showed that patients' preferences for type of consultation diverge, with some patients expressing strong preference for digital consultation and others for face-to-face consultation [49]. Factors such as duration of appointment, time of day, access to equipment, difficulty with activities, comorbidity, and travel costs have been identified as predictors of preference [49]. Preferences for participating in clinical decision-making also diverge, with some patients wanting to take part in the clinical decisions regarding their treatment, and others being more content to leave decisions to their physiotherapist [50]. Patients' preferences can also influence both choice of treatment and rehabilitation outcome [51], underscoring the importance of identifying patients who prefer digital consultation to a face-to-face encounter.

Physiotherapists in a recent survey have expressed that at least some of the needs of complex patients, e.g. those with communication difficulties or unclear diagnosis, were perceived to be better assessed in a face-to face consultation [52]. Virtual assessments alone could lead to inadequate examination and possible safety issues, or subsequent ineffective interventions. At the same time, the responding physiotherapists identified benefits with the environment in which the digital consultation takes place, and perceived virtual consultations as providing easy access and convenience for patients, giving patients flexibility of their appointments [52]. Another recent survey [53] found that lack of physical contact when working through telehealth could be a barrier for accurate and effective diagnosis and management, and that less than half (42%) of allied health clinicians surveyed believed telehealth was as effective as face-to-face care.

Our review indicates that there is not enough evidence for the validity of digital diagnosis even for well-defined patient populations. Diagnosis is obviously at the core of the physiotherapy assessment, and if the certainty of evidence is low for digital diagnosis, the overall physiotherapy assessment becomes a less viable alternative. A study evaluating digital paediatric physiotherapy consultations concluded that digital consultations have a place in physiotherapy, but are not suitable for all patients [54]. It is essential to investigate which patients benefit the most from digital consultations while also considering patients' preferences and needs [55]. According to Russel and colleagues [25], the patients shared more information when assessed face-to-face, which the authors believed enabled further clinical reasoning. This may also facilitate a person-centred approach in which the patient's narrative has a central role, an approach which is highly recommended in musculoskeletal pain care [54]. Further research is

warranted to examine whether digital and face-to-face assessments differ regarding the person-centred narrative and the partnership between patient and physiotherapist.

It is important to note that the digital tools available for clinical practice vary greatly, as does the suitability for digital assessment of different conditions and different patient populations [56]. To justify adoption of digital assessment as part of clinical practice, differences in validity and reliability in relation to standard face-to-face assessments should be negligible. There may be some common challenges for digital assessment and digital treatment; however, problems may differ and solutions should be tailored to the specific clinical situation. The currently very low certainty of evidence for digitally performed clinical tests, as well as for diagnosis and clinical management decisions based on digital examinations, imply that digital physiotherapy assessment should be used with caution.

Another relevant clinical issue is that the rapid development of technological solutions may be making the world more vulnerable in many ways [57]. In a digital society and health care, physiotherapists not only need to improve their own technical skills but also need to be comfortable acting as technical support and guiding patients as needed. Moreover, medical data may contain highly sensitive information and it is important to prevent medical information from being intercepted by malicious intruders. Ransomware has in recent years increased within health care in the western society [58]. Thus, both physiotherapists and patients using digital assessment must rely on IT providers to offer secure and reliable real-time video systems as well as systems to safely store their patients' health data [59, 60].

The varying validity, reliability and certainty of evidence shown in our review for different components of digital physiotherapy assessments suggests a need for future research to investigate optimisation of triaging to digital assessment or face-to-face assessment. The studies included in this review ranged in focus from specific lower limb or upper limb disorders to spinal pain. For each area, further research is needed.

A key advantage with digital assessment is that it can be accessed from almost anywhere. A transition to more digitally delivered health care could add to a reduction of the carbon footprint from health care due to the reduced need for travel [61] in sparsely populated areas or for specialised care [62, 63]. When evaluating digital interventions, it is important to address sustainable development in a broad perspective and a model has been suggested for evaluating ecological, economic, and social dimensions [64] with applications to physiotherapy [65]. Depending on context, digital health care may be cost-effective in primary health care for appropriate cases without increasing overall service use [66]. Concerning the social dimension, there still is a need to evaluate how digital assessment affects the interpersonal relations between the patient and the physiotherapist. Future research in the field of digital physiotherapy should relate outcomes to all three dimensions of sustainable development.

## Strengths and limitations

This is the first systematic review of digital physiotherapy assessment in which the GRADE system has been used to assess certainty of evidence, which is a main strength of the review in comparison to previous reviews. Other strengths include the composition of our review team comprising experienced musculoskeletal researchers most of whom also are clinically practicing physiotherapists, the comprehensive search strategy, and the use of two different quality appraisal tools. We believe our synthesised results presentation is a strength in that it indicates which types of assessments are best suited to digital solutions. Furthermore, we only included studies conducted in clinical populations with a range of musculoskeletal problems, increasing applicability and relevance of our review findings to clinical physiotherapy practice.

Limitations of the included studies include the risks of bias and concerns about applicability in four of the included studies, assessed with QUADAS-2, and concerns about applicability with all seven studies that investigated reliability, assessed with QAREL. The included studies were predominantly small and used more advanced digital systems with e.g. advanced motion analysis tools and e-Goniometers, than are normally available in clinical practice, reducing the applicability and generalisability of the results. Several studies were made in similar, limited settings, with the same researchers conducting several studies, also affecting applicability and generalisability of the review findings. These limitations affected our judgment of directness when determining certainty of evidence for validity and reliability of most of the components of the digital assessment. Our confidence in the findings was also frequently reduced due to concerns about precision, as many studies were small, had rather wide confidence intervals, or did not report confidence intervals at all. All of the included studies investigated only working-age adults which limits the generalisability concerning younger and older age groups.

A limitation with our review process is that, due to the heterogeneity of musculoskeletal conditions, tests used, and statistical measures, no meta-analysis was conducted. In our GRADE assessment, we did not formally assess the risk for publication bias due to the small and few studies identified; however, we did not see any indication of publication bias and estimated the risk to be low. We failed to identify any studies that investigated physiotherapist satisfaction, feasibility, adverse events or cost-effectiveness–all prespecified outcomes of interest for our review. A better understanding of both benefit/risk balance and cost/benefit balance of digital physiotherapy assessment is necessary before it would be advisable to implement this change in standard physiotherapy practice.

Assessing validity and reliability of physiotherapy assessment entails several challenges. First, the assessment is not a single test, but rather based on a package of many different tests, which varies depending on the patient, patient history and suspected diagnosis. Second, validity can be seen as an intermediate outcome for the effect of a diagnostic test on clinical outcomes, but there may not always be a direct linkage between validity of a diagnostic test and a clinical outcome. Third, applying the strength of evidence domains for studies of diagnostic tests is not as straightforward as for studies of intervention effectiveness.

For the risk of bias assessment, we chose two different checklists; the QUADAS-2 tool [11] and, because this tool does not address reliability [12], also the QAREL checklist. Because the items and focus of the two checklists were slightly different, the quality of the studies was rated differently for validity and reliability with the same study getting high quality rating on QUADAS-2, but a lesser rating on QAREL. For example, study limitations assessed with QAREL included raters' blinding, but in QUADAS-2 this is not specifically targeted. This also had consequences when it came to the GRADE assessment, where, for example, assessment of directness for the same study received a different grading for reliability and validity due to the different ratings in study population or raters in QUADAS-2 and QAREL.

## Conclusions

Evidence of variable certainty suggests that digital physiotherapy assessment may be compatible with face-to-face physiotherapy assessment, and that for most components, the two types of assessments may yield similar results in terms of validity and reliability. The highest validity was seen for extremity ROM assessment, PROMs, and clinical management, while the lowest validity was seen for posture assessment and clinical observations. In view of the variable validity and certainty of evidence, especially for central aspects of the physiotherapy assessment such as diagnosis, digital assessment should not completely replace face-to-face assessment, but may be relevant for patients who are likely to benefit from the accessibility and

convenience of remote access. The relatively high levels of agreement between the two types of assessment for several components, imply that digital physiotherapy assessment could be a relevant alternative to face-to-face assessment. It might be a beneficial way to ensure timely and environment-friendly access to care; however, limited access to digital tools, technical skills, and time-pressure are factors that might hinder widespread implementation of digital assessment. Furthermore, the patients' needs and preferences should be considered. While expressing high satisfaction with digital assessment, most patients seem to prefer face-to-face meetings when available. It is thus important to examine different perspectives concerning the possible benefits of digital assessment for patients, physiotherapists, and society. Moreover, the body of evidence for digital physiotherapy assessment is limited, the certainty of evidence varied from very low to high for the various assessment components investigated, and several outcomes of interest have not yet been examined. More research is needed before we can draw firmer conclusions on the benefits of digital physiotherapy assessment.

## Supporting information

**S1 File. PRISMA checklist.**
(DOCX)

**S2 File. Search strategy and results.**
(DOCX)

**S3 File. List of excluded studies.**
(DOCX)

**S4 File. Outcome tables.**
(DOCX)

## Acknowledgments

The authors wish to thank medical librarian Ida Stadig, Sahlgrenska University Hospital Medical Library, Gothenburg, Sweden, for her assistance in developing the search strategy and performing the database searches.

## Author Contributions

**Conceptualization:** Susanne Bernhardsson, Anette Larsson, Anna Bergenheim, Chan-Mei Ho-Henriksson, Annika Ekhammar, Elvira Lange, Maria E. H. Larsson, Lena Nordeman, Karin S. Samsson, Lena Bornhöft.

**Data curation:** Susanne Bernhardsson, Anette Larsson, Anna Bergenheim, Chan-Mei Ho-Henriksson, Annika Ekhammar, Elvira Lange, Maria E. H. Larsson, Lena Nordeman, Karin S. Samsson, Lena Bornhöft.

**Formal analysis:** Susanne Bernhardsson, Anette Larsson, Anna Bergenheim, Chan-Mei Ho-Henriksson, Annika Ekhammar, Elvira Lange, Maria E. H. Larsson, Lena Nordeman, Karin S. Samsson, Lena Bornhöft.

**Funding acquisition:** Maria E. H. Larsson, Lena Nordeman.

**Investigation:** Susanne Bernhardsson, Anette Larsson, Anna Bergenheim, Chan-Mei Ho-Henriksson, Annika Ekhammar, Elvira Lange, Maria E. H. Larsson, Lena Nordeman, Karin S. Samsson, Lena Bornhöft.

**Methodology:** Susanne Bernhardsson, Anette Larsson, Anna Bergenheim, Chan-Mei Ho-Henriksson, Annika Ekhammar, Elvira Lange, Maria E. H. Larsson, Lena Nordeman, Karin S. Samsson, Lena Bornhöft.

**Project administration:** Susanne Bernhardsson.

**Writing – original draft:** Susanne Bernhardsson, Anette Larsson, Anna Bergenheim, Chan-Mei Ho-Henriksson, Annika Ekhammar, Elvira Lange, Maria E. H. Larsson, Lena Nordeman, Karin S. Samsson, Lena Bornhöft.

**Writing – review & editing:** Susanne Bernhardsson, Anette Larsson, Anna Bergenheim, Chan-Mei Ho-Henriksson, Annika Ekhammar, Elvira Lange, Maria E. H. Larsson, Lena Nordeman, Karin S. Samsson, Lena Bornhöft.

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
