## [Decision Letter · Decision Letter 0]

13 Feb 2023

PONE-D-22-34676Digital physiotherapist assessment vs conventional face-to-face physiotherapist assessment of patients with musculoskeletal disorders: a systematic reviewPLOS ONE

Dear Dr. Susanne Bernhardsson,

Thank you for submitting your manuscript to PLOS ONE. After careful consideration, we feel that it has merit but does not fully meet PLOS ONE’s publication criteria as it currently stands. Therefore, we invite you to submit a revised version of the manuscript that addresses the points raised during the review process.

ACADEMIC EDITOR:Authors are requested to reply all the queries, raised by both the reviewers.

We look forward to receiving your revised manuscript.

Kind regards,

Priti Chaudhary, M.S.

Academic Editor

PLOS ONE

Journal Requirements:

2. Please include a copy of Table 5 which you refer to in your text on page 17.

Reviewers' comments:

Reviewer's Responses to Questions

**Comments to the Author**

1. Is the manuscript technically sound, and do the data support the conclusions?

Reviewer #1: Yes

Reviewer #2: Yes

2. Has the statistical analysis been performed appropriately and rigorously? 

Reviewer #1: N/A

Reviewer #2: Yes

3. Have the authors made all data underlying the findings in their manuscript fully available?

Reviewer #1: Yes

Reviewer #2: Yes

4. Is the manuscript presented in an intelligible fashion and written in standard English?

Reviewer #1: Yes

Reviewer #2: Yes

5. Review Comments to the Author

Reviewer #1: 1- Mani, S., Sharma, S., Omar, B., Paungmali, A., & Joseph, L. (2017). Validity and reliability of Internet-based physiotherapy assessment for musculoskeletal disorders: a systematic review. Journal of telemedicine and telecare, 23(3), 379-391. and

Grona, S. L., Bath, B., Busch, A., Rotter, T., Trask, C., & Harrison, E. (2018). Use of videoconferencing for physical therapy in people with musculoskeletal conditions: a systematic review. Journal of telemedicine and telecare, 24(5), 341-355.

You should explain the difference from studies. Especially your study is exactly the same as Suresh Manin's study. Even the studies you examine should clearly indicate your difference from that.

2- It would be more appropriate to say physiotherapy instead of physiotherapist in the title.

3- Your method part is a bit long and complicated, you can simplify and edit it according to the article below.

Yagiz, G., Akaras, E., Kubis, H. P., & Owen, J. A. (2022). The Effects of Resistance Training on Architecture and Volume of the Upper Extremity Muscles: A Systematic Review of Randomized Controlled Trials and Meta-Analyses. Applied Sciences, 12(3), 1593.

4- In the conclusion part, it should be emphasized in which evaluation parameters the remote evaluation is successful and in which it is unsuccessful. Make it clear that remote assessments are beneficial but not a substitute for face-to-face assessment.

5- Reload Figure 1 with better resolution.

Best wishes

Reviewer #2: Thank you for inviting me to review this article.

The authors did a good job. The manuscript was well-written, detailed and followed the PRISMA latest checklist for reporting systematic reviews. However, I have few observations to point out.

First, the reported location for "results of individual studies" item in the PRISMA 2020 checklist seems incorrect. Please check and make appropriate corrections.

Second, the flow diagram is blurred, I could not read it. Can another diagram be provided?

Third, line 525, page 25: This sentence "...validity of the digital tests varies with the validity of the reference tests..." negates your definition of validity. Validity is the extent of agreement between both tests, hence the validity of one cannot vary form the other. Kindly rephrase the sentence appropriately.

Finally, do you think the search strategy developed by the subject librarian needed a review by another librarian before conducting the search?

6. PLOS authors have the option to publish the peer review history of their article (what does this mean?). If published, this will include your full peer review and any attached files.

Reviewer #1: **Yes: **Esedullah AKARAS

Reviewer #2: No

---

## [Author Response · Author response to Decision Letter 0]

18 Feb 2023

Please see attached file "Response to reviewers".

---

## [Editor Report · Decision Letter 1]

1 Mar 2023

Digital physiotherapy assessment vs conventional face-to-face physiotherapy assessment of patients with musculoskeletal disorders: a systematic review

PONE-D-22-34676R1

Dear Dr. Susanne Bernhardsson,

We’re pleased to inform you that your manuscript has been judged scientifically suitable for publication and will be formally accepted for publication once it meets all outstanding technical requirements.

Kind regards,

Priti Chaudhary, M.S.

Academic Editor

PLOS ONE
---

## [Editor Report · Acceptance letter]

8 Mar 2023

PONE-D-22-34676R1 

Digital physiotherapy assessment vs conventional face-to-face physiotherapy assessment of patients with musculoskeletal disorders: a systematic review 

Dear Dr. Bernhardsson:

I'm pleased to inform you that your manuscript has been deemed suitable for publication in PLOS ONE. Congratulations! Your manuscript is now with our production department. 

Kind regards, 

on behalf of

Dr. Priti Chaudhary 

Academic Editor

PLOS ONE